# Dissociating the Effects of Light at Night from Circadian Misalignment in a Neurodevelopmental Disorder Mouse Model Using Ultradian Light–Dark Cycles

**DOI:** 10.3390/clockssleep7030048

**Published:** 2025-09-15

**Authors:** Sophia Anne Marie B. Villanueva, Huei-Bin Wang, Kyle Nguyen-Ngo, Caihan Tony Chen, Gemma Stark, Gene D. Block, Cristina A. Ghiani, Christopher S. Colwell

**Affiliations:** 1Integrative Biology and Physiology, University of California Los Angeles, Los Angeles, CA 90095, USA; svillanueva0131@g.ucla.edu (S.A.M.B.V.); tonycch417@g.ucla.edu (C.T.C.); gemmaestark@gmail.com (G.S.); gdblock@mednet.ucla.edu (G.D.B.); 2Department of Psychiatry & Biobehavioral Sciences, David Geffen School of Medicine, University of California Los Angeles, Los Angeles, CA 90095, USA; hueibinwang@gmail.com (H.-B.W.); kylenn210@g.ucla.edu (K.N.-N.); 3Department of Pathology & Laboratory Medicine, David Geffen School of Medicine, University of California Los Angeles, Los Angeles, CA 90095, USA

**Keywords:** autism spectrum disorders, circadian rhythms, *Cntnap2* KO, locomotor activity, neurodevelopmental disorders, T7 lighting, ultradian lighting

## Abstract

Individuals with neurodevelopmental disorders (NDDs) often experience sleep disturbances and are frequently exposed to light during nighttime hours. Our previous studies using the *Contactin-associated protein-like 2* (*Cntnap2*) knockout (KO) mouse model of NDDs demonstrated that nighttime light exposure adversely affected behavioral measures. In this study, we exposed wild-type (WT) and *Cntnap2* KO mice to an ultradian lighting cycle (T7), which alternates 3.5 h of light and 3.5 h of darkness, hypothesizing that this lighting protocol would mimic the impact of nighttime light exposure seen in standard light–dark cycles with dim light at night (DLaN). However, adult WT and *Cntnap2* KO mice held under the T7 cycle did not show the increased grooming behavior or reduced social interaction observed in *Cntnap2* KO mice exposed to DLaN. The T7 cycle lengthened the circadian period and weakened the rhythm amplitude without abolishing rhythmicity in either genotype. Finally, opposite to DLaN, neither the T7 cycle nor constant darkness (DD) elicited an increase in cFos expression in the basolateral amygdala. These results demonstrate that the adverse effects of nighttime light exposure in an NDD model depend on the extent of the circadian disruption rather than light exposure alone, emphasizing the importance of circadian stability as a protective factor in NDDs.

## 1. Introduction

A significant proportion of individuals with neurodevelopmental disorders (NDDs) experience disturbances of their daily sleep–wake cycles. Common complaints include delayed bedtimes and frequent nighttime awakenings [1,2,3,4]. These difficulties are associated with challenging daytime behaviors for the impacted individuals [5,6] but also negatively affect the sleep and overall health of their parents and/or caregivers [7,8]. Additionally, altered sleep patterns may lead to increased nighttime exposure to light from electronic screens [9,10,11], which even in young people without NDDs has been shown to delay sleep onset [12,13,14,15,16].

This led us to investigate whether nighttime light exposure could disrupt circadian timing and exacerbate NDD-associated symptoms in a transgenic mouse line lacking the *Contactin-associated protein-like 2* (*Cntnap2*) gene. This gene encodes CASPR2, a member of the neurexin family of transmembrane proteins [17], with key roles in brain development, synapse formation and function. Mutations in the *CNTNAP2* gene predispose to NDDs, including autism spectrum disorders (ASD); in particular, individuals with biallelic mutations develop a recognizable syndrome (often called CASPR2 deficiency disorder) with early-onset epilepsy, severe language impairment, intellectual disabilities, and autistic features in a substantial subset [18,19,20,21,22]. Mouse models also show epilepsy and autistic-like deficits [23,24]. Importantly for the present study, prior work has shown that *Cntnap2* KO mice show disruptions in circadian locomotor activity [25,26] and EEG-defined sleep patterns [27]. The validity of the *Cntnap2* KO mouse model is supported by intervention studies showing that: the FDA-approved drug risperidone reduces repetitive behaviors [23], administration of oxytocin enhances social behaviors [24], treatment with melatonin improves sleep–wake rhythms [26].

We have previously shown that exposing *Cntnap2* KO mice to dim light at night (DLaN) exacerbated the already abnormal locomotor activity and altered the neural activity in the suprachiasmatic nucleus (SCN), as well as the rhythms in clock gene expression as measured by PER2::LUC [26]. Notably, DLaN elicited excessive grooming behavior otherwise not observed in these mutants during the night [26,28]. These aberrant effects were mitigated by shifting to a long-wavelength (red) light [28]. An unresolved question is whether the detrimental effects of nighttime light exposure are due to light’s direct impact on behaviors or to the circadian disruption driven by DLaN. One approach to investigate this issue is to place the animals in an ultradian light–dark (LD) cycle that falls outside the circadian clock’s entrainment range. Prior work by Hattar and colleagues employed an ultradian 3.5 h light/3.5 h dark (T7) cycle to pinpoint the effects of light alone [29,30,31]. They reported that while the T7 cycle lengthens the circadian period, it does not disrupt internal rhythmicity of the SCN or cause arrhythmicity in sleep architecture and body temperature. The T7 cycle, however, did induce mood disturbances without increasing anxiety, suggesting that nighttime light alone can affect mood.

In the present study, we examined the impact of ultradian light exposure on wild-type (WT) and *Cntnap2* KO mice housed under T7 conditions (LD 3.5/3.5) in comparison to LD, constant darkness (DD), and DLaN. The impact of the T7 light cycle was assessed on social interactions using the three-chamber test, repetitive grooming behavior, and locomotor rhythms via passive infrared (PIR) detection. Finally, the effects of the T7 lighting on the expression of the immediate early gene cFos were investigated in the basal lateral amygdala (BLA), a region implicated in repetitive behaviors, since two weeks’ exposure to DLaN significantly increased the expression of this neural activity marker in this brain region in both the WT and mutants [28].

## 2. Results

Social deficits, a key symptom of NDD, have been reported in both juvenile [23] and adult [26,27,28,29] *Cntnap2* KO mice, and are exacerbated by exposure to DLaN [26,28]. To untangle the direct effects of exposure to light at night on behavior from those elicited by the circadian disruption, social interactions were assessed in WT and *Cntnap2* KO mice held for two weeks under one of four lighting conditions: LD, DLaN, T7, and DD (Figure 1). Mice in LD or DLaN, but not under T7 and DD, are synchronized to the environment, whilst those in DD exhibit a free-running rhythm driven by the endogenous clock.

The animals were tested in the three-chamber social arena during the active phase between Zeitgeber time (ZT) 17 and 19 if in LD or DLaN, or their subjective night (Circadian time (CT) 17–19) if in DD or T7 (Figure 1). The time spent in the chamber with either the novel mouse or the inanimate object was measured (Figure 2A,B) to determine their individual preference, and the resulting data analyzed by two-way ANOVA (Table 1). As previously reported [24,28], both the WT and mutants were significantly impacted by DLaN, with the WT exhibiting a significant reduction in the time spent with the novel mouse (*p* = 0.006), while neither T7 (*p* = 0.400) nor DD (*p* = 0.257) altered their social behavior compared to their counterpart in LD (Figure 2A,B). Similarly, in the *Cntnap2* KO mice, DLaN amplified the social deficits by reducing the time spent with the novel mouse (*p* = 0.002) as compared to the mutants in LD, whilst this effect was not observed in mice held in T7 (*p* = 0.879) or DD (*p* = 0.986) (Figure 2A). In comparison to the WT, the *Cntnap2* KO mice exhibited fewer social interactions regardless of the lighting conditions (LD: *p* < 0.001, DLaN: *p* < 0.001, T7: *p* = 0.049, and DD: *p* = 0.033); nonetheless, only DLaN significantly lessened sociability in both genotypes (Figure 2A,B).

Another hallmark symptom of NDDs is repetitive behavior, which is recapitulated in the *Cntnap2* mutants in the form of excessive grooming during the day [23] and night [26,28]. The time spent grooming was measured during the night (ZT 16–18) or subjective night (CT 16–18) (Figure 1). Significant effects of genotype and lighting cycle, as well as a significant interaction between the two factors, were revealed by two-way ANOVA (Table 1). In general, the WT mice spent a limited amount of time grooming (Figure 2C), which was not altered by DLaN (*p* = 0.750) or T7 (*p* = 0.339). The *Cntnap2* KO exposed to DLaN exhibited a dramatic increase in grooming (*p* < 0.001), but a similar augmentation was not seen under T7 (*p* = 0.907) (Figure 2C) as compared to mutants in LD. Despite the mutants exhibiting more grooming under each lighting condition (DLaN, *p* < 0.001; T7, *p* = 0.001; DD, *p* < 0.01) as compared to their WT counterparts, the aberrant effects of DLaN were significantly greater in comparison to the other three *Cntnap2* KO groups (Figure 2C).

The *Cntnap2* KO mouse model exhibits altered diurnal activity rhythms, including reduced nighttime activity [25,26,27,28]. To further characterize the effects of nighttime exposure to light, we assessed locomotor activity rhythms using passive infrared (PIR) sensors under each lighting condition. At least 10 consecutive days of activity data were collected per animal, and diurnal and circadian parameters were derived. As shown by the representative actograms (Figure 3A), both WT and KO mice exhibited rhythms with periods longer than 24 h when held on the T7 cycles (Table 2). While not a focus of the present study, we observed that both genotypes showed longer free-running periods when assessed by PIR rather than by wheel-running (Appendix A; see also [32]). Main effects of genotype and/or lighting conditions were identified by two-way ANOVA on total activity, period length, rhythm power, and onset variability, along with a significant genotype × lighting interaction for both period and power (Table 2).

In WT mice, DLaN significantly reduced the power of the rhythms (*p* = 0.001), with no significant changes in other parameters; the T7 cycles had a broader impact, significantly increasing the period (*p* < 0.001) and onset variability (*p* < 0.001) whilst also reducing rhythm power (*p* < 0.001) relative to those in LD and/or DD (Figure 3B and Table 2). In the *Cntnap2* KO mice, DLaN did not alter the power of the rhythms (*p* = 0.07) but significantly increased the onset variability (*p* = 0.037; Figure 3B), while T7 illumination significantly lengthened the free-running period (*p* < 0.001) and increased the variability of the activity onset (*p* < 0.001; Table 2 and Figure 3B) as compared to their counterparts in LD or DD. Direct comparisons between genotypes revealed significant differences in total activity (*p* = 0.035), rhythm power (*p* = 0.015), and onset variability (*p* = 0.004) with no changes in period (*p* = 0.644) under LD, as well as in onset variability (*p* < 0.001) and period (*p* < 0.001) under DLaN and T7, respectively. The effect of genotype was less dramatic under DD conditions, where a robust reduction in power (*p* < 0.001) was detected (Figure 3B and Table 2) with no changes in the other parameters.

Finally, sex-divergent effects were a striking feature of this dataset (Figure 4). While this variable did not seem to influence the social and grooming behaviors (Figure 4A), significant effects were observed for total activity, period, power, and onset variation (three-way ANOVA with genotype, lighting cycles, and sex as factors, Figure 4B and Table 3).

Overall, the *Cntnap2* KO mice exhibited weaker locomotor activity rhythms, revealing robust genotype-specific effects of T7 illumination. Furthermore, sex-dependent differences emerged as a prominent feature, underscoring the importance of including both sexes in circadian rhythm studies involving ASD models.

Evidence suggests that the BLA, located in the temporal lobe of the cerebral cortex, is a critical mediator of aberrant repetitive behaviors [33,34]; in addition, we have previously shown that DLaN elicits increased expression of cFos, a marker of neural activity, in a population of glutamatergic neurons in this nucleus [28] in both WT and *Cntnap2* KO mice, which correlated with the observed behavioral changes. Therefore, we examined its expression in the BLA of mice held in the four lighting conditions (Figure 5). A significant effect of the lighting cycles, but not of genotype, was present (Table 4), with both WT and *Cntnap2* KO mice exposed to DLaN displaying a significant increase in the number of cFos-positive cells as compared to their counterparts in LD (*p* = 0.0002 and *p* < 0.0001, respectively; Figure 5 and Table 4), whilst neither the T7 cycle nor DD altered the number of cFos positive cells. In general, both WT and mutants held in T7 or in DD presented with a lower number of positive cells as compared to their counterparts in LD, with a significant difference between WT in LD and DD (*p* = 0.0283 WT in DD vs. WT in LD). Notwithstanding the well-documented effect of LD and DLaN on cFos expression in the BLA in our previous work [28], we included these groups in the present study for a more accurate evaluation of the effects observed, hence the small sample size. These findings are consistent with the behavioral evidence suggesting that nighttime light exposure by itself is not sufficient to activate BLA neurons and elicit the aberrant effects observed.

## 3. Discussion

Patients with NDDs, including ASD, often present with delayed sleep onset, fragmented sleep, and blunted circadian rhythms [1,2,3,4,5,6]. We hypothesize that a circadian disruption not only contributes to the core symptoms of NDDs but also renders affected individuals more vulnerable to its adverse effects. This vulnerability suggests a potential benefit from circadian-based therapeutic interventions.

To investigate this hypothesis, we employed the *Cntnap2* KO mouse, a well-established model of ASD and related NDDs [35]. Our prior work has shown that these mutants exhibit abnormal activity and sleep rhythms, altered neural activity in the SCN, and exaggerated sensitivity to environmental circadian disruption. Furthermore, exposure to DLaN triggers repetitive behaviors, otherwise absent in the *Cntnap2* KO during the night, aggravates their social deficits [26,28], and worsens the altered SCN neural activity and PER2::LUC-driven rhythms [26]. Notably, administration of melatonin restores rhythmicity and mitigates behavioral abnormalities, with the greatest improvement observed in animals displaying the most robust circadian rhythms.

In the current study, we examined the effects of a non-24 h ultradian lighting schedule—specifically, a T7 cycle (3.5 h light/3.5 h dark)—on behavior and activity rhythms in *Cntnap2* KO and WT mice. The T7 cycle exposes the animals to light during their active phase each circadian cycle, and both genotypes exhibited free-running rhythms with periods longer than 24 h. Consistent with previous reports [23,26], the mutation alone triggered reduced social interactions but not abnormal grooming during the night/active phase; strikingly, *Cntnap2* KO mice under T7 did not display a worsening of their behavioral deficits, in contrast to the response to DLaN (Figure 2). Prior work by the Hattar group showed that T7 exposure induces depression-like behaviors (e.g., reduced sucrose preference, increased immobility in the forced swim test) and impairs memory performance in tasks such as the Morris water maze and novel object recognition [29,31]. These impairments were mostly assessed during the subjective day; however, recent findings from Fuchs and colleagues [36] suggest that behavioral outcomes under T7 conditions are phase-dependent, with mood and memory impairments emerging only when mice are tested during their subjective night.

Although prior reports indicated that the T7 cycle does not overtly disrupt the central circadian clock—based on SCN gene expression, body temperature rhythms, and sleep architecture [29,37]—our high-resolution locomotor activity analysis revealed subtle but important alterations (Figure 3). After two weeks in T7, both WT and KO mice maintained their rhythmicity, albeit with significant lengthening of the period and increased variability in the activity onset, consistent with prior work [29,37]. Notably, T7 lighting also reduced the power of the rhythms and increased onset variability of the mice relative to DD. Moreover, we detected robust sex differences in these circadian parameters (Figure 4), emphasizing the importance of sex as a biological variable.

Our findings highlight a key dissociation: while T7 alters the activity rhythms, it does not elicit the same behavioral changes seen with DLaN in the *Cntnap2* KO mice, suggesting that light at night alone is insufficient to exacerbate autistic-like behaviors; rather, an interaction between light exposure and circadian misalignment may be required (see also [38]). The differential behavioral outcomes in mice under DLaN vs. T7 provide a valuable framework to dissect mechanistic underpinnings of light-induced pathology. Thus, we conclude that while the T7 lighting does produce some disturbances, DLaN exposure is more behaviorally perturbing.

To identify the relevant neural circuits, we build on previous findings that melanopsin-expressing intrinsically photosensitive retinal ganglion cells (ipRGCs) are necessary for DLaN-induced effects, since *Opn4^DTA^* mice lacking these cells do not show DLaN-induced changes in locomotor activity [37]. Additionally, DLaN increases cFos expression in glutamatergic neurons of the BLA, a region implicated in social behavior and repetitive grooming [28,33,34]. In contrast, T7 exposure did not elicit a cFos response in the BLA (Figure 5), nor did it exacerbate grooming or social deficits. Future studies using chemogenetic silencing or targeted lesions of BLA cell populations will be instrumental in testing its causal role in mediating the behavioral impact of DLaN.

A key innovation of this study is the use of an ultradian T7 lighting schedule to dissociate the direct effects of nighttime light exposure from the circadian disruption it often induces. Prior research has established that DLaN elicits, alters, and/or amplifies behavioral and physiological abnormalities in both WT and *Cntnap2* KO mice. However, it has remained unclear whether these adverse outcomes result from the exposure to light per se or from the misalignment of internal circadian rhythms. By implementing the T7 cycle—comprising 3.5 h of light and dark intervals that fall outside the range of circadian entrainment—we provide evidence that the deleterious effects of DLaN are contingent on circadian misalignment. Despite frequent nocturnal light exposure, mutant mice under the T7 cycle maintained their rhythmicity and did not exhibit excessive repetitive behavior, worsening of the pre-existing social deficits, or elevated cFos expression in the amygdala triggered by DLaN. These findings support the “two-hit” hypothesis of NDDs: a genetic predisposition (e.g., *Cntnap2* mutation) primes the neural circuits for dysfunctionality, which can be further amplified by an environmental disturbance or stressor. We propose that circadian disruption can serve as the environmental stressor or “second hit” for those who are vulnerable.

Several limitations should be acknowledged. First, we employed 250 lx illumination for the T7 cycles which was consistent with earlier work. It would be interesting to compare the behavioral outcomes driven by T7 using different light intensities (10–250 lx). Second, we measured both the autistic-like behaviors and cFos expression at a fixed phase (ZT/CT 16–18) and did not carry out a time-series analysis. Third, while our activity rhythm analysis had a sufficient sample size to detect sex differences, this was not the case for the autistic-like behavioral measurements, and the lack of statistically significant sex differences in these measures should be viewed cautiously (Figure 4).

Our findings advance the field by refining the mechanistic understanding of how light at night affects behavior and by introducing ultradian lighting as a novel paradigm for dissecting circadian versus non-circadian influences on neural function in a disease model. Most importantly, they have translational implications for individuals with NDDs, including ASD, who are frequently exposed to light at nighttime from electronic devices and/or artificial lighting. Our results demonstrate that the adverse behavioral and neural effects of light at night in a genetically susceptible model are contingent on circadian disruption, rather than exposure to light alone. This distinction emphasizes the importance of maintaining circadian rhythmicity as a protective factor against environmental challenges in individuals at risk because of their genetic make-up. Interventions aimed at supporting circadian health—such as structured light exposure schedules, consistent sleep–wake timing, or the use of circadian-stabilizing agents like melatonin—may help mitigate the impact of nocturnal light exposure in individuals with NDDs. Furthermore, the ultradian lighting paradigm used here provides a mechanistically informative model for testing the efficacy of such circadian-targeted interventions without introducing the confounding effects of arrhythmicity. Ultimately, these findings support the development of chronobiology-informed strategies to improve behavioral outcomes and quality of life in individuals with NDDs.

## 4. Materials and Methods

### 4.1. Animals and Experimental Groups

All animal procedures were performed in accordance with the UCLA animal care committee’s regulations. A total of 111 adult (3–4 months of age) mice (55 WT and 56 mutants) were used for these experiments, with a mixed number of males and females in each experimental group. *Cntnap2^tm1Pele^* mutant mice [39] back-crossed into the C57BL/6J background strain were acquired (B6.129(Cg)-*Cntnap2*^tm2Pele^/J; stock #017482; RRID:IMSR_JAX:017482) from The Jackson Laboratory (Bar Harbor, ME, USA). Mice of the WT C57BL/6J and of the *Cntnap2* null mutant (KO) strain were from our breeding colony maintained in an approved facility of the Division of Laboratory Animal Medicine at the University of California, Los Angeles (UCLA). The mice had free access to food and water and were entrained in a 12 h:12 h LD cycle for two weeks before being randomly assigned to one of the following groups: (A) continuing in the normal LD cycle, (B) releasing in constant darkness (DD), and exposure to (C) DLaN (10 lx illumination during lights off) or (D) the T7 ultradian cycle (3.5 h light: 3.5 h dark), for two additional weeks (Figure 1). We used a two-week exposure to DLaN based on prior data showing that this duration is enough to alter our behavioral measures [25]. The regular light was set at 250 lx as measured at the floor of the animal holding chambers while DLaN was measured at 10 lx (see Appendix A for more information about the lighting).

### 4.2. Behavioral Tests

A cohort of 40 WT and 40 *Cntnap2* KO (3–4 months-old) was used to test social and grooming behavior after 2 weeks’ exposure to one of four lighting conditions (LD, DLaN, T7, or DD; Figure 1). On the 14th day, the animals were placed in a novel arena for 10 min to measure grooming and exploration behavior during their active or subjective active phase (ZT 17–19 or CT 17–19). After habituation to the arena, they were tested using the three-chamber sociability protocol [40]. In this test, mice are free to roam an arena with three chambers. The central chamber remains empty while the flanking chambers contain an up-turned metal-grid pencil cup: one is kept empty to be the novel object (O), while an age- and sex-matched WT stranger mouse (S) is placed in the other. The stranger mice had previously been habituated to the cup for 3 × 15 min sessions. To test for social preference, mice were presented with the choice between O and S. The time that the tested mouse spent investigating the object or the stranger mouse was determined. Social preference was determined by comparing the dwell times of the tested mice in the two chambers, and the social index calculated using the following formula: time with S/time with O, where a higher value indicates greater social preference. The three-chamber test was performed under dim red light (<2 lx at arena level) during the active phase (ZT 17–19 or CT 17–19). Video recordings were captured using a Sony CMOS video camera, supplemented with infra-red lighting, connected to a video-capture card (Adlink Technology Inc., Irvine, CA, USA) on a Dell Optiplex computer system. Mice were automatically tracked using the ANY-maze software (V. 7.48; Stoelting, Wood Dale, IL, USA). Grooming behaviors were manually scored and averaged post hoc by two independent observers masked to the experimental groups.

### 4.3. Cage Conditions and Activity

Mice were housed individually to monitor and collect locomotor activity rhythms using a top-mounted passive infrared (PIR) motion detector reporting to a VitalView data-recording system (Mini Mitter, Bend, OR, USA) over a period of at least two weeks (Figure 1). The cages were placed in circadian controlled chambers where an environment with a temperature range of 65–75° and humidity levels of 30–40% was maintained. Detected movements were recorded in 3 min bins, and at least 10 days of data were averaged for analysis using the Clocklab program (Actimetrics, Wilmette, IL, USA). The strength of the rhythms was determined from the amplitude of the χ^2^ periodogram at 24 h, to produce the rhythm power (%V) normalized to the number of data points examined. Other locomotor activity parameters were calculated using the Clocklab program. Under LD or DLaN conditions, the time of lights OFF was defined as ZT 12. Under DD or T7 conditions, the time of activity onset was defined at circadian time (CT) 12.

### 4.4. Immunofluorescence

At the end of the two-week exposure to one of the four lighting cycles (Figure 1), mice were anesthetized with isoflurane (30–32%) at a specific time during the night (ZT 18) or subjective night (CT 18) and transcardially perfused with phosphate-buffered saline (PBS, 0.1 M, pH 7.4) containing 4% (*w*/*v*) paraformaldehyde (Sigma, St. Louis, MO, USA). The brains were rapidly dissected out, post-fixed overnight in 4% PFA at 4 °C, and cryoprotected in 15% sucrose. Coronal sections (50 μm) were obtained using a cryostat (Leica, Buffalo Grove, IL, USA), collected sequentially, and paired along the anterior–posterior axis before further processing. Immunohistochemistry was performed as previously described [28,41] Briefly, free-floating coronal sections containing the BLA were blocked for 1 h at room temperature (1% BSA, 0.3% Triton X-100, 10% normal donkey serum in 1xPBS) and then incubated overnight at 4 °C with a rabbit polyclonal antiserum against cFos (1:1000, clone 9F6, Cell Signaling Technology, Danvers, MA, USA) followed by a Cy3-conjugated donkey-anti-rabbit secondary antibody (Jackson ImmunoResearch Laboratories, Bar Harbor, ME, USA). Sections were mounted and coverslips applied with Vectashield mounting medium containing DAPI (4′-6-diamidino-2-phenylinodole; Vector Laboratories, Burlingame, CA, USA) and then visualized on a Zeiss AxioImager M2 microscope (Zeiss, Thornwood, NY, USA) equipped with a motorized stage, a monochromatic camera AxioCamMRm (Zeiss, Thornwood, NY, USA), and the ApoTome imaging system.

### 4.5. cFos-Positive Cell Counting in the Basolateral Amygdala (BLA)

The BLA was visualized using the DAPI nuclear staining and Z-stack images (7 μm interval, 40 images) of both the left and right BLA acquired with a 20× objective using the Zeiss Zen digital imaging software (version 3.12). The cells immunopositive for cFos were counted with the aid of the Zen software tool ‘marker’ in three to five consecutive sections by three observers masked to the genotype and experimental groups. The values obtained from the left and right BLA of each slice were averaged, and the means of the three to five slices were then averaged to obtain one value per animal. Data are presented as the mean ± S.D. of three to seventeen animals per light treatment.

### 4.6. Statistical Analysis

Data analysis was performed using Prism (Version 10.5.0; GraphPad Software, La Jolla, CA, USA) or SigmaPlot (version 16, SYSTAT Software, San Jose, CA, USA). Two-way analysis of variance (ANOVA) followed by the Holm–Šidák test for multiple comparisons or the Bonferroni’s multiple comparisons test with genotype and light cycle conditions as factors was used to analyze the impact of the different lighting conditions on social interactions and grooming behavior, parameters of the locomotor activity rhythms, and the number of cFos-positive cells in the BLA (Figure 2, Figure 3 and Figure 5; Table 1, Table 2 and Table 4). Three-way ANOVA with sex, genotype, and light cycle conditions as factors followed by the Holm–Šidák test for multiple comparisons was used to analyze the effect of sex on social and grooming behavior as well as some parameters of locomotor activity rhythms (Table 3; Figure 4). Values are reported as the mean ± standard deviation (SD). Differences were determined to be significant if *p* < 0.05.

## Figures and Tables

**Figure 1 clockssleep-07-00048-f001:**
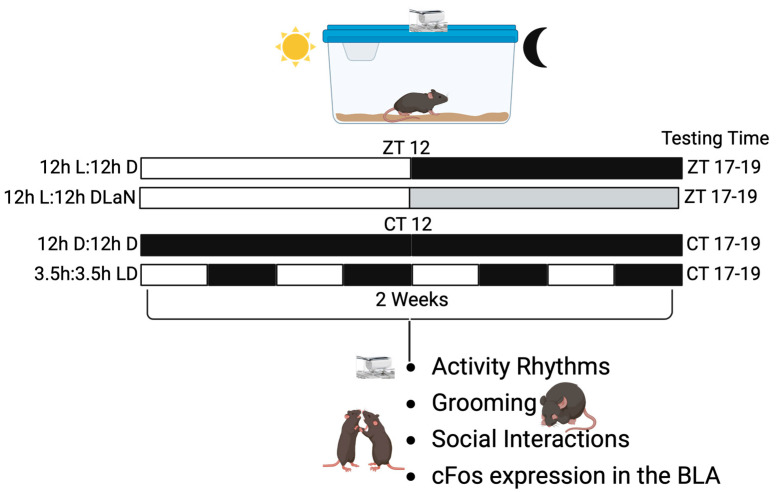
Experimental design. WT and *Contactin-associated protein-like 2* (*Cntnap2*) knockout (KO) mice (both sexes) were entrained to a 12 h:12 h light–dark (LD) cycle then randomly assigned to one of four lighting conditions: 12 h:12 h LD; constant darkness (DD); dim light at night (DLaN): 12 h of bright light at 250 lx followed by 12 h of dim (10 lx) light; and a ultradian LD cycle consisting of 3.5 h of light followed by 3.5 h of dark (T7). Rhythms in cage activity were recorded using passive infrared sensors for at least two weeks, then the mice were tested during the dark phase/night between Zeitgeber time (ZT) or Circadian time (CT) 17 and 19 for grooming behavior and social interactions. ZT 12 was defined as the time of lights OFF for mice synchronized to external lighting conditions (LD and DLaN), and CT 12 as the time of activity onset for free-running animals (DD and T7). A separate cohort of mice, held in the same four lighting conditions, was euthanized at ZT18 or CT18, and the brains processed for immunofluorescence with an antibody against cFos. BLA = basolateral amygdala. Created in https://BioRender.com.

**Figure 2 clockssleep-07-00048-f002:**
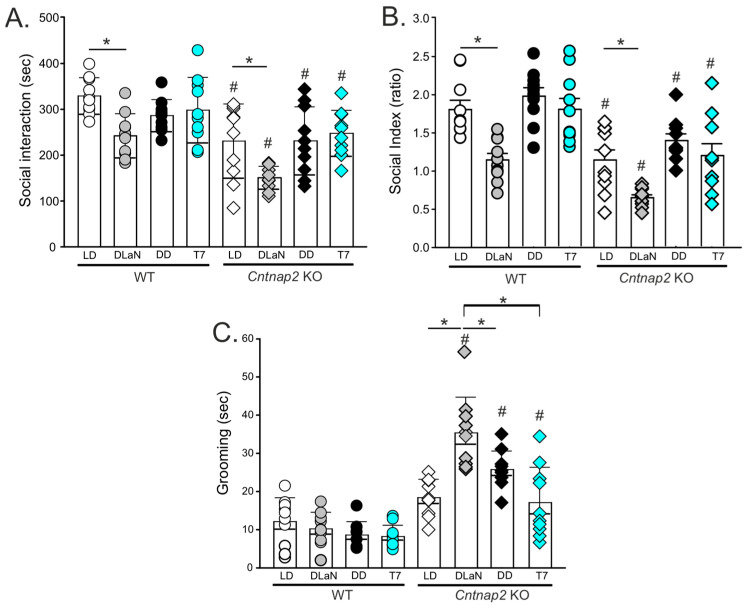
T7 lighting does not mimic Dim Light at Night (DLaN)-driven changes in autistic-like behavior in the *Cntnap2* KO mice. A mixed number of male and female mice from both genotypes were held under one of four lighting conditions: LD, DLaN, DD, or a T7 cycle (*n* = 10 mice per genotype) for two weeks. Behavioral assessments were conducted during the active phase between ZT 17 and 19 or CT 17 and 19. (**A**) Social behavior was evaluated by analyzing the time actively spent by the testing mouse interacting with the novel stranger mouse. The stranger and testing mice were matched for age, sex, and genotype. DLaN significantly reduced social interactions in both WT and *Cntnap2* KO mice. In contrast, the mice held in T7 did not show changes in social interactions when compared to their counterparts in LD or DD. The time spent interacting with the novel mouse was significantly reduced in the mutants regardless of the light cycle as compared to their respective WT counterparts. (**B**) Likewise, the mutants presented with a lower social preference index, i.e., decreased interest in the stranger mouse, as compared to WT in the same lighting condition, but only DLaN impacted sociability in both WT and *Cntnap2* KO mice compared to their counterpart in LD. (**C**) Grooming was assessed in a novel arena between ZT 17 and 19 or CT 17 and 19. The time spent grooming was not altered by any of the lighting conditions in WT mice. In contrast, DLaN, but not T7, triggered excessive grooming behavior in the *Cntnap2* KO mice compared to mutants held in LD or DD conditions. Compared to WT, the *Cntnap2* KO exhibited increased grooming under DLaN, DD, and T7 conditions. Bar graphs show the means ± SD with overlaid the values from individual WT and *Cntnap2* KO mice held in each of the four lighting conditions. Data were analyzed using a two-way ANOVA with genotype and lighting cycle as variables (see Table 1) followed by the Holm–Šidák multiple comparisons test. Asterisks indicate significant differences (* *p* < 0.05) between light cycles—same genotype, whilst crosshatches indicate significant differences (# *p* < 0.05) between genotypes—same lighting condition.

**Figure 3 clockssleep-07-00048-f003:**
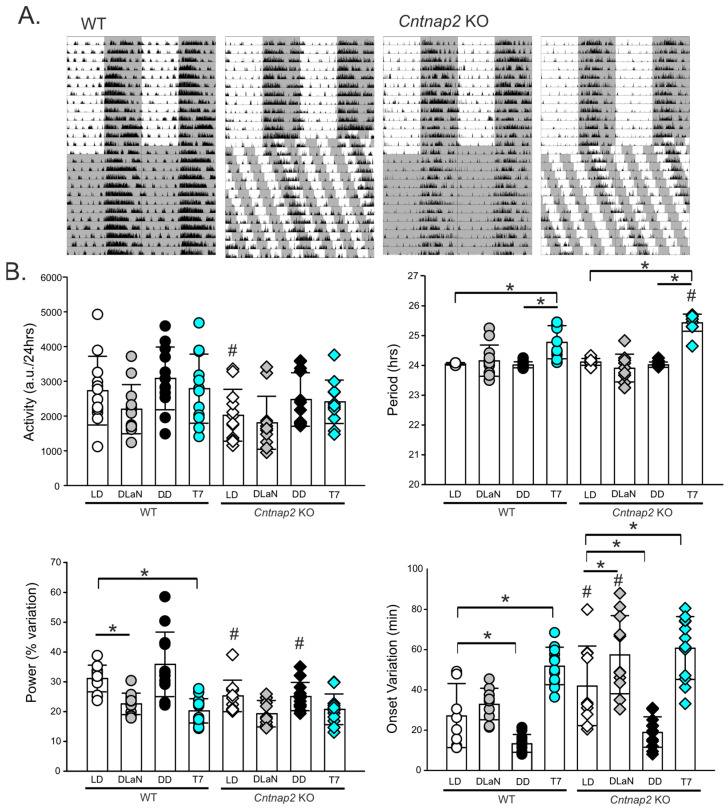
The T7 cycle lengthens the period and weakens, but does not abolish, rhythmicity in locomotor activity. A mixed number of male and female mice of both genotypes were entrained for two weeks in LD and then held for two additional weeks under one of four lighting conditions: LD, DLaN, DD, and a T7 cycle (*n* = 12 mice per genotype). (**A**) Examples of actograms of daily rhythms in cage activity of WT and *Cntnap2* KO mice held in DD and in a T7 cycle. The activity levels in the actograms were normalized to the same scale (85% of the maximum of the most active individual). Each row represents two consecutive days, and the second day is repeated at the beginning of the next row. The gray shading represents the time of darkness. (**B**) Properties of the daily activity rhythms (10-day recordings) in each of the four lighting conditions: LD, DLaN, DD, and T7. In WT mice, the T7 cycle lengthens the period, reduces the power, and increases the onset variation of the activity rhythms. In the *Cntnap2* KO mice, the T7 lighting also increases the period and onset variation. Compared to WT, the KO mice in LD exhibit reduced activity levels, reduced power, and increased onset variation (Table 2). Bar graphs show the means ± SD with overlaid the values from individual WT and *Cntnap2* KO mice. Data were analyzed by two-way ANOVA with genotype and treatment as variables (see Table 2) followed by the Holm–Šidák multiple comparisons test. The asterisks indicate significant differences (* *p* < 0.05) between lighting cycles within genotype, and the crosshatches indicate significant differences (# *p* < 0.05) between the two genotypes—same lighting cycle.

**Figure 4 clockssleep-07-00048-f004:**
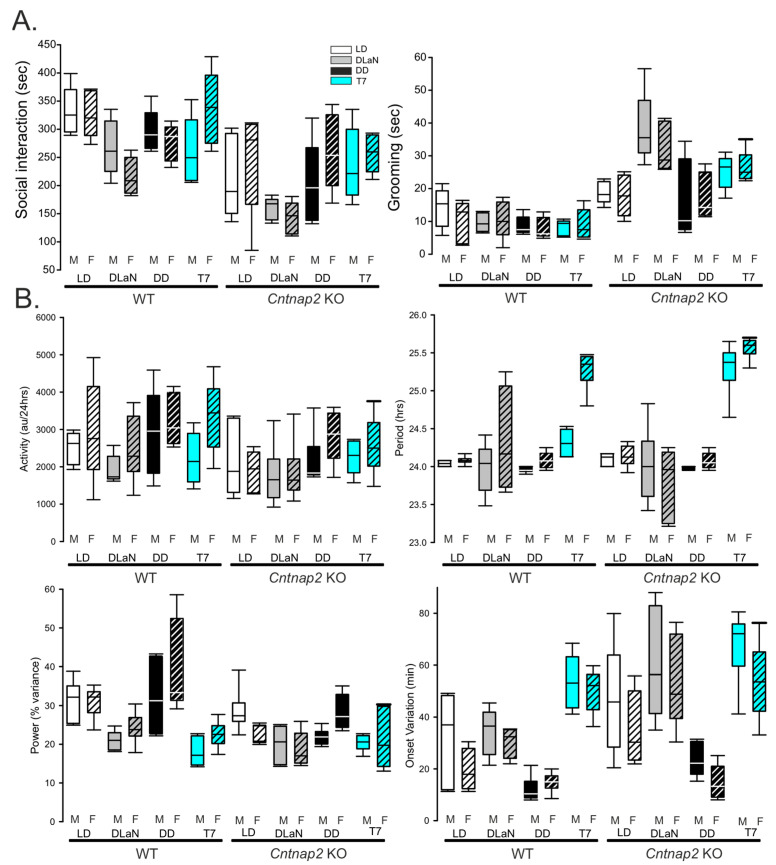
Effect of sex on autistic-like behaviors and activity rhythms in WT and *Cntnap2* KO mice. The behavioral measures described above were also segregated by sex. (**A**) Autistic-like behaviors did not vary with sex. Data were analyzed by three-way ANOVA with genotype, lighting cycles, and sex as variables followed by the Holm–Šidák multiple comparisons test. Significant main effects of genotype and lighting cycles (Social Interaction: F(1,79) = 35.360; *p* < 0.001; F(3,79) = 9.264; *p* < 0.001); Grooming: F(1,79) = 63.660; *p* < 0.001; F(3,79) = 4.445; *p* < 0.001) but not of sex or their interaction, were observed; however, with a sample size of *n* = 5 per group, per sex, these datasets are not strongly powered to evaluate sex differences, and the negative findings should be viewed with caution. (**B**) Significant main effects of sex were observed on total activity, period, and onset variation (three-way ANOVA with genotype, lighting cycles, and sex as variables followed by the Holm–Šidák multiple comparisons test; Table 3), with no significant sex differences between groups. The effects of T7 lighting on circadian rhythms did not differ between males (M) and females (F) of the same genotype. Box plots show the statistical values for each group, where the boundary of the box closest to zero indicates the 25th percentile, the line within the box marks the median, and the boundary of the box farthest from zero indicates the 75th percentile. The whiskers above and below the box indicate the 90th and 10th percentiles. The sample size was *n* = 6 per group per sex for these measures.

**Figure 5 clockssleep-07-00048-f005:**
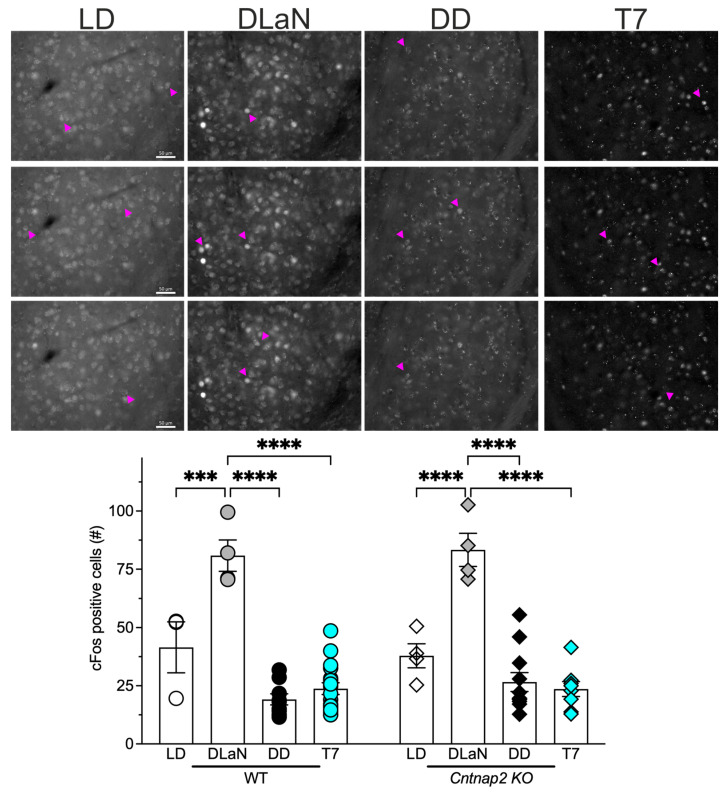
DLaN, but not T7 lighting, evoked an increase in cFos expression in the basolateral amygdala (BLA) of WT and *Cntnap2* KO mice. Mice from each genotype, with a mixed number of males and females, were entrained to a 12 h:12 h LD cycle for two weeks and then held under one of four lighting conditions: LD, DLaN, DD, and a T7 cycle for two additional weeks (*n* = 3–17 mice per group). Mice were euthanized and perfused at ZT 18 or CT 18, the brains rapidly dissected and processed for immunofluorescence. (**Upper Panels**) Representative alternate images of cFos expression in the mutants’ BLA. Arrowheads: cFos-positive cells; scale bar = 50 µm. (**Lower Panel**) The number of cFos-positive cells was determined bilaterally in the BLA. The values from three to five consecutive coronal sections/animal (both left and right hemispheres) were averaged to obtain one value per animal (*n* = 3 to 17). DLaN elicited a robust increase in the number of cFos-expressing cells (arrowheads) in both the WT and the mutants; conversely, no induction was observed in any of the other groups. No genotypic differences were observed. Bar graphs show the means ± SD with overlaid the values from the individual WT and *Cntnap2* KO mice in each of the four lighting conditions: LD, DLaN, DD, and T7. Data were analyzed by two-way ANOVA with genotype and treatment as variables (see Table 4) followed by the Bonferroni’s multiple comparisons test. The asterisks indicate significant differences in lighting conditions within genotype: *** *p* = 0.0002; **** *p* < 0.0001.

**Table 1 clockssleep-07-00048-t001:** Impact of different lighting cycle conditions on social interactions and grooming in WT and *Cntnap2* KO mice. Values are shown as the averages of the time measured in seconds per each animal per activity ± SD (*n* = 10 mice per genotype). The social preference ratio was calculated using the following formula: time with stranger mouse/time with the object, and a higher value indicates greater social preference. Data were analyzed by two-way ANOVA with genotype and lighting cycle conditions as factors, followed by the Holm–Šidák test for multiple comparisons. Asterisks indicate significant differences between lighting cycles within genotype, and crosshatches indicate genotypic differences—same lighting cycle. Bold values indicate significance, alpha = 0.05.

	WT				*Cntnap2* KO				Two-Way ANOVA		
	LD	DLaN	DD	T7	LD	DLaN	DD	T7	Genotype	Lighting	Interaction
Social intera-ctions (s)	329 ± 40	**242 ± 48 ***	286 ± 35	292 ± 51	**231 ± 81 ^#^**	**151 ± 25 *^#^**	**231 ± 74 ^#^**	**248 ± 50 ^#^**	**F_(1,79)_ = 33.628; *p* < 0.001**	**F_(3,79)_ = 8.811; *p* < 0.001**	F_(3,79)_ = 0.906; *p* = 0.443
Social pre-ference (ratio)	1.8 ± 0.4	**1.1 ± 0.3 ***	1.9 ± 0.3	1.8 ± 0.4	**1.1 ± 0.4 ^#^**	**0.6 ± 0.4 *^#^**	**1.3 ± 0.3 ^#^**	**1.2 ± 0.5 ^#^**	**F_(1,79)_ = 51.562; *p* < 0.001**	**F_(3,79)_ = 17.101; *p* < 0.001**	F_(3,79)_ = 0.181; *p* = 0.909
Grooming (s)	12.1 ± 6.2	10.2 ± 4.3	8.6 ± 3.5	8.2 ± 2.9	18.4 ± 4.8	**35.3 ± 9.3 *^#^**	**25.7 ± 4.8 ^#^**	**17.1 ± 9.2** ^#^	**F_(1,79)_ = 66.571; *p* < 0.001**	**F_(3,79)_ = 4.648; *p* = 0.005**	**F_(3,79)_ = 4.512; *p* = 0.006**

**Table 2 clockssleep-07-00048-t002:** Impact of different light cycle conditions on locomotor activity rhythm parameters in WT and *Cntnap2* KO mice. Values are shown as the averages ± SD (*n* = 12 mice per genotype). Data were analyzed by two-way ANOVA, with genotype and light cycle conditions as factors, followed by the Holm–Sidäk test for multiple comparisons. Asterisks indicate significant differences between light cycles within genotype, and crosshatches indicate genotypic differences—same lighting cycle. * *p* < 0.001 vs. WT or *Cntnap2* KO in LD or DD; # *p* < 0.05 vs. WT in the same light cycle. Bold values indicate significance, alpha = 0.05.

	WT				*Cntnap2* KO				Two-Way ANOVA		
	LD	DLaN	DD	T7	LD	DLaN	DD	T7	Genotype	Lighting	Interactions
Total activity (a.u./24 h)	2730 ± 988	2180 ± 707	3084 ± 903	2788 ± 995	**2023 ± 747 ^#^**	1806 ± 763	2477 ± 769	2410 ± 624	**F_(1,95)_ = 9.648; *p* = 0.003**	**F_(3, 95)_ = 3.993; *p* = 0.010**	F_(3,95)_ = 0.235; *p* = 0.872
Period (h)	24.0 ± 0.05	24.2 ± 0.52	24.0 ± 0.10	**24.8 ± 0.5 ***	24.1 ± 0.11	23.9 ± 0.5	24.0 ± 0.09	**25.4 ± 0.29*^#^**	F_(1,95)_ = 2.937; *p* = 0.090	**F_(3, 95)_ = 58.80; *p* < 0.001**	**F_(3,95)_ = 7.647; *p* < 0.001**
Power (% variance)	31.1 ± 4.5	**22.6 ± 3.6 ***	35.8 ± 6.8	**20.3 ± 4.1 ***	**25.3 ± 5.3 ^#^**	19.3 ± 4.4	**25.1 ± 4.7 ^#^**	20.8 ± 5.1	**F_(1,95)_ = 17.20; *p* < 0.001**	**F_(3, 95)_ = 18.60; *p* < 0.001**	**F_(3,95)_ = 4.093; *p* = 0.009**
Fragmentation # bouts/24 h	10.9 ± 4.1	11.7 ± 5.7	10.2 ± 1.8	10.5 ± 1.4	10.3 ± 4.8	12.4 ± 4.4	11.1 ± 2.6	11.8 ± 1.7	F_(1,95)_ = 0.588; *p* = 0.445	F_(3, 95)_ = 0.844; *p* = 0.473	F_(3,95)_ = 0.281; *p* = 0.839
Onset variability (min)	28.0 ± 16.6	32.7 ± 8.2	**13.4 ± 4.4 ***	**51.8 ± 9.3 ***	**43.0 ± 20 ^#^**	**57.5 ± 19.4*^#^**	**19.5 ± 7.6 ***	**60.8 ± 15.6 ***	**F_(1,95)_ = 20.938; *p* < 0.001**	**F_(3, 95)_ = 37.007; *p* < 0.001**	F_(3,95)_ = 2.105; *p* = 0.106

**Table 3 clockssleep-07-00048-t003:** Sex affects the impact of different lighting cycle conditions on some parameters of locomotor activity rhythms in WT and *Cntnap2* KO mice. Data were analyzed by three-way ANOVA with sex, genotype, and lighting cycle conditions as factors; only the values for the interactions among the three values are shown (*n* = 6 mice per sex per genotype). Bold values indicate significance, alpha = 0.05.

	Sex	Genotype	Lighting	Interactions
Total activity (a.u./24 h)	**F_(1,95)_ = 5.853; *p* = 0.018**	**F_(1,95)_ = 10.01; *p* = 0.002**	**F_(3,95)_ = 4.144; *p* = 0.009**	F_(3,95)_ = 0.648; *p* = 0.586
Period (h)	**F_(1.95)_ = 13.18; *p* < 0.001**	**F_(1,95)_ = 4.239; *p* = 0.043**	**F_(3,95)_ = 84.86; *p* < 0.001**	**F_(3,95)_ = 2.577; *p* = 0.060**
Power (% variance)	F_(1 95)_ = 2.441; *p* = 0.122	**F_(1,95)_ = 18.94; *p* < 0.001**	**F_(3,95)_ = 20.49; *p* < 0.001**	F_(3,95)_ = 0.256; *p* = 0.857
Fragmentation (#bouts/24 h)	F_(1,95)_ = 1.276; *p* = 0.262	F_(1,95)_ = 0.585; *p* = 0.447	F_(3,95)_ = 0.840; *p* = 0.476	F_(3,95)_ = 0.072; *p* = 0.975
Onset variation (min)	**F_(1,89)_ = 8.022; *p* = 0.006**	**F_(1 89)_ = 16.508; *p* < 0.001**	**F_(3,89)_ = 38.041; *p* < 0.001**	F_(3,89)_ = 0.171; *p* = 0.915

**Table 4 clockssleep-07-00048-t004:** DLaN, but not T7, elicits a significant increase in the number of cFos-positive cells in the basolateral amygdala (BLA) of WT and *Cntnap2* KO mice. Cells were counted in z-stack images of both the left and right BLA from three to five sections/animal, and these values averaged to obtain one value per section and then one value per animal. Data were analyzed by two-way ANOVA with genotype and lighting cycles as factors followed by the Bonferroni’s multiple comparisons test. Values are shown as the averages ± SD of 3 to 17 mice per genotype and lighting cycle, alpha = 0.05. Bold values indicate significance. Asterisks indicate significant differences between lighting cycles within genotype; no genotypic differences were observed. * *p* < 0.05 or *** *p* < 0.001 vs. WT or *Cntnap2* KO in LD or DD.

	Genotype		Two-Way ANOVA		
Lighting Cycle	WT	*Cntnap2* KO	Genotype	Lighting	Interaction
LD	41.5 ± 18.9	37.9 ± 10.3	*F*_(1,52)_ = 0.1959; *p* = 0.6599	***F*_(3,52)_ = 60.63; *p* < 0.0001**	*F*_(3,52)_ = 0.5808; *p* = 0.6302
DLaN	**80.8 ± 13.5 *****	**83.3 ± 14.2 *****			
DD	**19.1 ± 7.09 ***	26.6 ± 13.4			
T7	23.8 ± 10.4	23.6 ± 9.05			

## Data Availability

Data are available upon request. Upon acceptance, we will deposit data in the publicly archived Dryad database (https://datadryad.org/, accessed on 11 September 2025).

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
