# Peer review of "Dissociating the Effects of Light at Night from Circadian Misalignment in a Neurodevelopmental Disorder Mouse Model Using Ultradian Light–Dark Cycles"

_2624-5175, 2025, doi:10.3390/clockssleep7030048_

Round 1

Reviewer 1 Report

Comments and Suggestions for Authors

Previously, the team conducted a study to examine the effect of dim light at night (DLaN), on the social and autistic behaviors of Cntmap2 KO mice, as a model of neurodevelopmental disorders (NDD), and found that DLaN strongly exacerbates NDD associated behavioral symptoms. In the current study, to delineate whether the action mechanism of DLaN is the direct light effect or circadian disruption, the authors used a T7 short light:dark cycle to which mice cannot entrain. Interestingly, unlike DLaN, the T7 cycle did not exacerbate their social interaction score nor excessive grooming behavior in Cntnap2 KO mice. T7 cycle lengthened the periods and weakened circadian behavior rhythms in both WT and Cntmap2 KO mice. C-Fos staining of baso-lateral amygdala (BLA) revealed that DLaN, but not T7, increased cFos expression in both WT and Cntnap2 KO mice. Overall, these results provide interesting insights into the mechanism of clock disruption that exacerbate autistic behaviors of an NDD mouse model. The following issues should be addressed.

  1. The T7 ultradian cycle didn’t replicate the behavioral and neurobiological changes observed from DLaN, although they have severe impact on circadian rhythms. These results are consistent with the notion that dim light exposure at night and light-induced neuronal activation may be a key factor leading to behavioral exacerbation in Cntnap2 KO mice. But the author’s conclusion is the opposite, which requires clarification.

Are the authors suggesting that during T7, mice were exposed to light during their subjective night, which should be considered as nighttime light exposure? The authors’ interpretation is that light at night alone is insufficient to exacerbate autistic-like behaviors in Cntnap2 KO mice. These findings suggest that more severe or specific forms of circadian misalignment—beyond the perturbations induced by T7—may be necessary to trigger such behavioral changes? But I am confused since DLaN caused severe behavior changes which is dim light only at night time and presumably it is causing more severe circadian misalignment than T7 cycle?

Overall the studies together show that DLaN-type light exposure is more behaviorally disruptive than T7, but how that can lead to the conclusion that circadian misalignment is the key factor should be better explained.

  1. Line101: Although T7 and DD behaviors can be described as not synchronized, they are still fundamentally different. It’s recommended to distinguish T7 (disruption) and DD (free running driven by the endogenous clock).

  1. Figure 5. C-Fos staining in BLA: Scale bar is missing. Please indicate with arrows for a couple of examples of c-Fos positive cells that were counted for quantification.

  1. Supplemental Fig. 2: Free-running period was measured in WT and Cntnap2 KO mice under constant darkness, with either running wheels or passive infrared (PIR) sensors. Please provide wheel-running circadian parameters as in Fig 3.

Author Response

  1. The T7 ultradian cycle didn’t replicate the behavioral and neurobiological changes observed from DLaN, although they have severe impact on circadian rhythms. These results are consistent with the notion that dim light exposure at night and light-induced neuronal activation may be a key factor leading to behavioral exacerbation in Cntnap2 KO mice. But the author’s conclusion is the opposite, which requires clarification.

Are the authors suggesting that during T7, mice were exposed to light during their subjective night, which should be considered as nighttime light exposure? The authors’ interpretation is that light at night alone is insufficient to exacerbate autistic-like behaviors in Cntnap2 KO mice. These findings suggest that more severe or specific forms of circadian misalignment—beyond the perturbations induced by T7—may be necessary to trigger such behavioral changes? But I am confused since DLaN caused severe behavior changes which is dim light only at night time and presumably it is causing more severe circadian misalignment than T7 cycle?

Overall the studies together show that DLaN-type light exposure is more behaviorally disruptive than T7, but how that can lead to the conclusion that circadian misalignment is the key factor should be better explained.

Thank you for the insightful comments.  We agree with the reviewer’s statement that DLAN exposure is behaviourally more disruptive than the T7.  We also agree that we may have over-stated our point that circadian misalignment is the key difference.  We are now more cautious in our interpretations. 

  1. Line101: Although T7 and DD behaviors can be described as not synchronized, they are still fundamentally different. It’s recommended to distinguish T7 (disruption) and DD (free running driven by the endogenous clock).

We agree and have altered this line accordingly (lines 85-87).

  1. Figure 5. C-Fos staining in BLA: Scale bar is missing. Please indicate with arrows for a couple of examples of c-Fos positive cells that were counted for quantification.

Our apologies, the figure was reduced in size so much that the scale bars in the left panels were not visible. We enlarged the figures and the scale bars in the left panels (LD) more visible and added harrowheads.

  1. Supplemental Fig. 2: Free-running period was measured in WT and Cntnap2 KO mice under constant darkness, with either running wheels or passive infrared (PIR) sensors. Please provide wheel-running circadian parameters as in Fig 3.

We have added the data that we have to supplemental figure (now suppl Fig1). This was not a central part of the data set but something that came out of our analysis and we thought that the readers may find interesting. 

Reviewer 2 Report

Comments and Suggestions for Authors
  1. The manuscript focuses on the CNTNAP2 gene, which is indeed implicated in a broad spectrum of disorders, including but not limited to Autism Spectrum Disorder (ASD). However, the current version places disproportionate emphasis on neurodevelopmental disorders, which may inadvertently obscure the broader biological and clinical significance of CNTNAP2. It is recommended that the authors remove ASD-related content to maintain clarity and focus.

See:Toma C, Pierce KD, Shaw AD, Heath A, Mitchell PB, et al. (2018) Comprehensive cross-disorder analyses of CNTNAP2 suggest it is unlikely to be a primary risk gene for psychiatric disorders. PLOS Genetics 14(12): e1007535. https://doi.org/10.1371/journal.pgen.1007535

2.  Materials and Methods section,the content in lines 447–450 and 487–490 is redundant and should be simplified.

3. Please provide the model of the video recorder and the version of the software used for behavioral analysis.

4. The Results section needs to be more concise. Please remove redundant expressions and avoid overlap with the Discussion section. For instance, lines 94–97 contain interpretations that belong to the Discussion, while lines 97–99 describe methodological details and should be moved to the Materials and Methods section.

5. Figures are missing captions.

6. The Discussion section repeats many points already mentioned in the Results. Please condense the Discussion and focus on interpretation rather than reiterating the data.

Author Response

REVIEWER 2

The manuscript focuses on the CNTNAP2 gene, which is indeed implicated in a broad spectrum of disorders, including but not limited to Autism Spectrum Disorder (ASD). However, the current version places disproportionate emphasis on neurodevelopmental disorders, which may inadvertently obscure the broader biological and clinical significance of CNTNAP2. It is recommended that the authors remove ASD-related content to maintain clarity and focus.

We agree with the reviewer’s point that Cntnap2 mutations have been associated with a range of disorders including ASD.  When both copies of CNTNAP2 are disrupted, individuals develop a recognizable neurodevelopmental syndrome (often called CASPR2-deficiency) with early-onset epilepsy, severe language impairment, ID, and autistic features in a substantial subset (≈41% in one recent cohort). We feel that this is a robust gene–disorder link, though it’s syndromic rather than “idiopathic ASD.”

The evidence is mixed for heterozygous variants in Cntnap2 being linked to idiopathic ASD. Large case–control and exome datasets have not found convincing enrichment of heterozygous rare variants in CNTNAP2 among people with ASD, and common-variant signals have been inconsistent or failed to replicate at genome-wide significance.

Regardless, the Cntnap2 KO has become one of the most common mouse models used to study ASD, because they reproduce behavioral, neurobiological, or genetic features relevant to the disorder. For example, the Cntnap2 KO mice show deficits in social interactions and repetitive behaviors as well as well described deficits in sleep and circadian rhythms.

We provide a number of citations that support the use of the Cntnap2 KO as a model to explore some to the symptoms relating to ASD.  We are not making the argument the Cntnap2 mutations are causal for ASD nor is this critical for our study. 

Materials and Methods section: the content in lines 447–450 and 487–490 is redundant and should be simplified.

We have simplified these two paragraphs, and lines 487-490 have been removed as suggested.

Please provide the model of the video recorder and the version of the software used for behavioral analysis.

We apologies for the omission, a Sony CMOS video recorder and ANY-maze (version 7.48) behavioral analysis Software were used and now indicate this in the Methods.

The Results section needs to be more concise. Please remove redundant expressions and avoid overlap with the Discussion section. For instance, lines 94–97 contain interpretations that belong to the Discussion, while lines 97–99 describe methodological details and should be moved to the Materials and Methods section.

We have worked to tighten up the results section, however some of the concepts in these paragraphs are necessary to explain the rationale behind the study design.

Figures are missing captions.

Our apologies, when we transferred the text to the journal template, the figure legends somehow ended up mixed with the main text.  We have corrected this error. 

The Discussion section repeats many points already mentioned in the Results. Please condense the Discussion and focus on interpretation rather than reiterating the data.

We have tightened up the discussion.

Reviewer 3 Report

Comments and Suggestions for Authors

This is a very interesting study investigating the effects of different lighting cycles on the behavior and amygdala neural activity in wildtype and Cntnap2 KO mice. The data suggest a dissociation of the effects of light exposure during night and circadian disruption on autistic-like behavior and amygdala activity. The study is well performed, the data of high interest and the manuscript is well written. However, I recommend to adress several questions/comments when revising the manuscript.

Comments/questions:

(1) Line 42: Consider introducing the gene Cntnap2. In this context, is there any work on heterozygous Cntnap2 KO mice?

(2) Figure 1: If the figure was made with the help of BioRender (it looks like this), the authors have to mention this (see the conditions for using BioRender). Consider indicating the testing time(s) in the figure.

(3) Consider enlarging the size of all figures.

(4) Line 91: The brains were processes...,  Line 92: ...amygdala...

(5) Line 103ff: In my opinion, there are more factors that have to be considered for the statistical analyses. a critical factor is the explored subject (inanimated object vs. novel mouse). Further, sex is an important biological factor.

I suggest that Figure 2 present the novel mouse exploration time. It is important to interpret this in the context of object exploration time. The quality of the effect differs greatly depending on whether both social and object exploration are similarly affected, or if only social exploration is impacted. The authors may also consider calculating an exploration ratio to better illustrate this comparison.

(6) Figure 2: Is there no statistical difference in grooming between the LD and the DLaN condition in Cntnap2 mice?

(7) Results: in general, it would be helpful for the reader if the quality of an effect is also mentioned (increase, decrease,...).

(8) Line 215 "The study was powered to evaluate potential sex differences in activity rhythms" vs. line 256ff "with a sample size of n=5 per group, per sex, these data sets are not powered to evaluate sex differences". These statements are from the same experiment, correct?

(9) Figure 5: Please comment the low sample size for the LD and DLaN groups. In the behavioral analyses, there are much more animals in these groups.

(10) Line 279: "...; in addition, we have previously shown that DLaN elicits increase cFos expression in a population of glutamatergic neurons in this nucleus [29]." I did not find this result in the given citation. 

 (11) Line 351ff: I only partially understand this "two-hits" idea. Is disrupted sleep-wake cycle now a symptom or a cause ("hit") of NDD? If "Cntnap2 KO mice exhibit disrupted activity and sleep rhythms" during LD (line 345), it is a symptom (or behavioral endophenotype) and not a "hit", correct? 

(12) Line 358ff: "Strikingly, Cntnap2 KO mice did not show in creased grooming or reduced social interaction under T7, in contrast to their behavioral responses to DLaN (Fig. 2A,B), while, consistent with previous [22, 27, 25], these behavioral abnormalities persisted under standard LD conditions". In the figure & result section, I only find a significant reduction of social interaction while grooming is not affected by the genotype.

(13) Line 376ff: Moreover, we detected robust sex differences in these circadian parameters (Fig. 4), emphasizing the need to consider sex as a biological variable when evaluating circadian disruption." This argumentation is a bit strange for me. In general, sex should always be considered as a biological variable. If not, sex differences cannot be detected. And also, if not sex differences are/were detected, sex should be considered as a biological variable.

(14) Line 413: "Of course, given that we have four lighting conditions and two genotypes, a time series sampling every 4 hrs would have required hundreds of additional mice." Why "of course"? Autistic-like behavior such as reduced social interaction and increased grooming could also be continuously observed under homecage conditions, i.e. without disturbing the animals. Then, one could easily evaluate the time courses of these behaviors by a within-subject design and would not need hundreds of mice. Instead of cFOS, one could record from amygdala and also do this continously.

(15) Line 416: What qualifies to be "powered to detect sex difference"?

(16) Methods, lines 469-470: Here, the formula for a social preference ratio is given. Why not presenting the ratios?

Author Response

REVIEWER 3

This is a very interesting study investigating the effects of different lighting cycles on the behavior and amygdala neural activity in wildtype and Cntnap2 KO mice. The data suggest a dissociation of the effects of light exposure during night and circadian disruption on autistic-like behavior and amygdala activity. The study is well performed, the data of high interest and the manuscript is well written. However, I recommend to address several questions/comments when revising the manuscript.

Comments/questions:

Line 42: Consider introducing the gene Cntnap2. In this context, is there any work on heterozygous Cntnap2 KO mice?

We agree with the reviewer and have added information about the Cntnap2 gene (lines 41-55).

Cntnap2 heterozygous mice (+/−) don’t show robust ASD-like behaviors under standard conditions. In the classic three-chamber test, wild-type and heterozygotes behave similarly, while homozygous knockouts show the social deficits. Even without classic ASD-like behavior, heterozygous mice can show white-matter/myelination changes and motor-coordination deficits (e.g., grid-walking), indicating that haploinsufficiency affects some circuits, just not enough to generate full ASD-like behaviors by itself (see 10.3389/fnins.2023.1100121). For neurodevelopmental research, homozygote Cntnap2 KO mice are commonly used.  

Figure 1: If the figure was made with the help of BioRender (it looks like this), the authors have to mention this (see the conditions for using BioRender). Consider indicating the testing time(s) in the figure.

We added the testing times and acknowledged BioRender in the figure legend.

Consider enlarging the size of all figures.

We agree with the reviewer and have enlarged the figures.

Line 9: The brains were processes...,  Line 92: ...amygdala...

Thank you, we have corrected these errors. 

Line 103ff: In my opinion, there are more factors that have to be considered for the statistical analyses. a critical factor is the explored subject (inanimated object vs. novel mouse). Further, sex is an important biological factor.

We now report the ratio of time spent exploring the novel mouse over the time spent exploring the object. 

The effects of sex on grooming and social interactions are shown in Fig 4 top panels.

I suggest that Figure 2 present the novel mouse exploration time. It is important to interpret this in the context of object exploration time. The quality of the effect differs greatly depending on whether both social and object exploration are similarly affected, or if only social exploration is impacted. The authors may also consider calculating an exploration ratio to better illustrate this comparison.

We now report the ratio of time spent exploring the novel mouse over the time spent exploring the object (see Fig. 2B; Table 1).

Figure 2: Is there no statistical difference in grooming between the LD and the DLaN condition in Cntnap2 mice?

Thank you for pointing this out, we have added the asterisk to the figure. DLaN does increase grooming in the mutant mice (also reported in table 2).

Results: in general, it would be helpful for the reader if the quality of an effect is also mentioned (increase, decrease,...).

We have revised the text to emphasize the directionality of the effects. 

Line 215 "The study was powered to evaluate potential sex differences in activity rhythms" vs. line 256ff "with a sample size of n=5 per group, per sex, these data sets are not powered to evaluate sex differences". These statements are from the same experiment, correct?

The statement on line 215 refers to the sample size for the activity rhythms, while line 256 (legend to fig 4) refers only to the data shown in panel A, see revised figure legend.

Figure 5: Please comment the low sample size for the LD and DLaN groups. In the behavioral analyses, there are much more animals in these groups.

We have previously published that 2 weeks exposure to DLaN elicits a significant increase in the number of cFos positive cells (Fig 5; Wang et al Neurobiol of Dis 2023; https://doi.org/10.1016/j.nbd.2022.105944).  For the present study, we added these two groups as control for the lack of effect of T7 and DD on this marker expression, instead of just citing the earlier work.  We have acknowledge this on lines 185-187.

Line 279: "...; in addition, we have previously shown that DLaN elicits increase cFos expression in a population of glutamatergic neurons in this nucleus [29]." I did not find this result in the given citation. 

Please see figure 5 in Wang et al 2023,(ref #28 in the revised version) https://doi.org/10.1016/j.nbd.2022.105944, also pasted here. A similar result was obtained in the peri-habenula area.

Line 351ff: I only partially understand this "two-hits" idea. Is disrupted sleep-wake cycle now a symptom or a cause ("hit") of NDD? If "Cntnap2 KO mice exhibit disrupted activity and sleep rhythms" during LD (line 345), it is a symptom (or behavioral endophenotype) and not a "hit", correct? 

Individuals with NDD carry genetic risk factors (hit 1) that primes the neural circuits to be vulnerable to environmental disturbance or stressor (hit 2). In our work, we have been providing evidence that disruptions to the circadian timing system (DLaN) can serve as stressor (hit 2).  We have reworked the discussion to clarify this point.  (lines 269-273)

Line 358ff: "Strikingly, Cntnap2 KO mice did not show increased grooming or reduced social interaction under T7, in contrast to their behavioral responses to DLaN (Fig. 2A,B), while, consistent with previous [22, 27, 25], these behavioral abnormalities persisted under standard LD conditions". In the figure & result section, I only find a significant reduction of social interaction while grooming is not affected by the genotype.

We have reworked this sentence as suggested.

Line 376ff: Moreover, we detected robust sex differences in these circadian parameters (Fig. 4), emphasizing the need to consider sex as a biological variable when evaluating circadian disruption." This argumentation is a bit strange for me. In general, sex should always be considered as a biological variable. If not, sex differences cannot be detected. And also, if not sex differences are/were detected, sex should be considered as a biological variable.

We agree that sex should always be considered as a biological variable.  We have reworked this sentence accordingly. (lines 278-281)

Line 413: "Of course, given that we have four lighting conditions and two genotypes, a time series sampling every 4 hrs would have required hundreds of additional mice." Why "of course"? Autistic-like behavior such as reduced social interaction and increased grooming could also be continuously observed under homecage conditions, i.e. without disturbing the animals. Then, one could easily evaluate the time courses of these behaviors by a within-subject design and would not need hundreds of mice. Instead of cFOS, one could record from amygdala and also do this continously.

We agree with the reviewer and are working on bringing such video analysis technology to the lab. In the meantime, we have modified this sentence. 

For sex-difference studies in Line 416: What qualifies to be "powered to detect sex difference"?

We typically power for genotype × sex interaction in a 2×2 design (Genotype: WT vs KO; Sex: ♀ vs ♂). Type I error (α).  For our power analysis, we typically use 0.05 for Type 1 error and power (1−β) commonly 0.80.  In practice, we find that we need 8-10 mice of each sex to determine whether sex differences exist. 

Methods, lines 469-470: Here, the formula for a social preference ratio is given. Why not presenting the ratios?

We now present the ratios as suggested (see Fig. 2B).
